# Preparation of TiH$_{1.924}$ nanodots by liquid-phase exfoliation for enhanced sonodynamic cancer therapy

Fei Gong[1], Liang Cheng[1✉], Nailin Yang[1], Yuehan Gong[1], Yanwen Ni[1], Shang Bai[1], Xianwen Wang [1], Muchao Chen[1], Qian Chen[1] & Zhuang Liu [1✉]

Metal hydrides have been rarely used in biomedicine. Herein, we fabricate titanium hydride (TiH$_{1.924}$) nanodots from its powder form via the liquid-phase exfoliation, and apply these metal hydride nanodots for effective cancer treatment. The liquid-phase exfoliation is an effective method to synthesize these metal hydride nanomaterials, and its efficiency is determined by the matching of surface energy between the solvent and the metal hydrides. The obtained TiH$_{1.924}$ nanodots can produce reactive oxygen species (ROS) under ultrasound, presenting a highly efficient sono-sensitizing effect. Meanwhile, TiH$_{1.924}$ nanodots with strong near-infrared (NIR) absorbance can serve as a robust photothermal agent. By using the mild photothermal effect to enhance intra-tumoral blood flow and improve tumor oxygenation, a remarkable synergistic therapeutic effect is achieved in the combined photothermal-sonodynamic therapy. Importantly, most of these TiH$_{1.924}$ nanodots can be cleared out from the body. This work presents the promises of functional metal hydride nanomaterials for biomedical applications.

[1] Institute of Functional Nano & Soft Materials (FUNSOM), Jiangsu Key Laboratory for Carbon-Based Functional Materials and Devices, Soochow University, Suzhou 215123, China. ✉email: lcheng2@suda.edu.cn; zliu@suda.edu.cn

Sonodynamic therapy (SDT) triggered by ultrasound (US) is a non-invasive therapeutic strategy that can be applied to treat deeply-seated tumors[1–4]. During SDT, sono-sensitizers are able to interact with surrounding oxygen and even water molecules to produce cytotoxic reactive oxygen species (ROS) to kill tumor cells[2,4–7]. However, the limitations of current sono-sensitizers have substantially hindered the extensive clinical applications of SDT. Traditional organic sono-sensitizers (e.g., photofrin[8], phthalocyanine[9], and chlorophyll derivative[10]), which are often derived from photo-sensitizers, often show photo-toxicity toward the skin[11,12]. The most representative paradigm of inorganic sono-sensitizers is semiconductor titanium dioxide ($TiO_2$)[13–15], whose quantum yield of US-triggered ROS generation, however, is relatively low due to the fast combination of the electron ($e^-$) and holes ($h^+$) ($50 \pm 30$ ns)[16,17].

Titanium hydride ($TiH_{1.924}$) has been explored for applications in hydrogen storage[18,19], and is frequently used as a foaming agent in the production of metallic foams (e.g., Zn, Al foams)[20,21], as well as a raw material for producing highly purified titanium and titanium alloys[22,23]. Considering the unique valence status of Ti (containing $Ti^0$, $Ti^{2+}$, $Ti^{3+}$, and $Ti^{4+}$) in $TiH_{1.924}$,[24] we hypothesize that it might be easily activated by external stimuli (e.g., light, ultrasound, and microwave) for applications in photo-catalysis and sono-catalysis[25,26]. However, nano-structured $TiH_{1.924}$ has not yet been synthesized to our best knowledge.

Liquid-phase exfoliation usually by sonicating bulk materials in appropriate solvents is a simple top-down route to produce various types of nanomaterials. In this work, we successfully exfoliate $TiH_{1.924}$ powder into ultrasmall nanodots via the liquid-phase exfoliation technology, and find that the surface energy plays an important role in the formation of the ultrasmall $TiH_{1.924}$ nanodots. In addition, this liquid-phase exfoliation is an effective method to synthesize various types of metal hydride nanomaterials (e.g., $TiH_{1.924}$, $ZrH_2$, $CaH_2$, and $HfH_{1.983}$). Taking $TiH_{1.924}$ nanodots for example, they have highly effective US-triggered ROS generation capability, which is superior to the sono-sensitizing effect of titanium dioxide ($TiO_2$), the classical inorganic sono-sensitizer, likely owing to the reduced bandgap in

$TiH_{1.924}$. Moreover, these black $TiH_{1.924}$ nanodots with strong near-infrared (NIR) absorption can also use as an excellent photothermal agent. Taking the advantage of mild photothermal effect to enhance intra-tumor blood flow and improve oxygen supply, a remarkably synergistic photothermal-sonodynamic therapeutic outcome has been achieved with $TiH_{1.924}$ nanodots (Fig. 1). In a mouse tumor model, the complete tumor eradication without recurrence is achieved after intravenous injection of $TiH_{1.924}$ nanodots and exposure of tumors to light and ultrasound, sequentially. Importantly, these $TiH_{1.924}$ nanodots with ultra-small sizes show efficient body excretion and no appreciable toxicity to the treated animals. This work highlights the potential of metal hydride nanomaterials as physical stimuli-triggered nanoagents for cancer treatment.

## Results

**Preparation and characterization of $TiH_{1.924}$ nanodots.** Liquid-phase exfoliation technology has been widely reported for the preparation of mono- or few-layered two-dimensional nanosheets[27–30]. In this work, we unexpectedly found that metal hydrides powder could be easily exfoliated into nanodots using liquid-phase exfoliation technology in the presence of appropriate solvents (Fig. 2a). Taking $TiH_{1.924}$ for example, we initially sonicated commercial $TiH_{1.924}$ powder in a number of exfoliation solvents (Fig. 2a, b). Among these 13 solvents including water, glycerol, dimethyl sulfoxide/N-methyl pyrrolidone (DMSO/NMP), DMSO, polyethylene glycol 200 (PEG 200), NMP, N,N-dimethylforma-mide (DMF/NMP), pyridine, DMF, acetonitrile, tetrahydrofuran (THF), ethanol and acetone, 6 solvents of them (water, glycerol, acetonitrile, THF, ethanol, and acetone) showed no exfoliation effects to the $TiH_{1.924}$ powder, while the other 7 solvents (DMSO/NMP, DMSO, PEG 200, NMP, DMF/NMP, pyridine, and DMF) could efficiently exfoliate the $TiH_{1.924}$ powder into small nano-particles (Fig. 2b). The different exfoliation results might be attributed to the surface energy of these solvents (Fig. 2c)[27,31–33]. For these 13 solvents, when the surface energy is too high ($H_2O$, 72.7; glycerol, 63.4 mJ m$^{-2}$) or too low (acetonitrile, 28.1; THF,

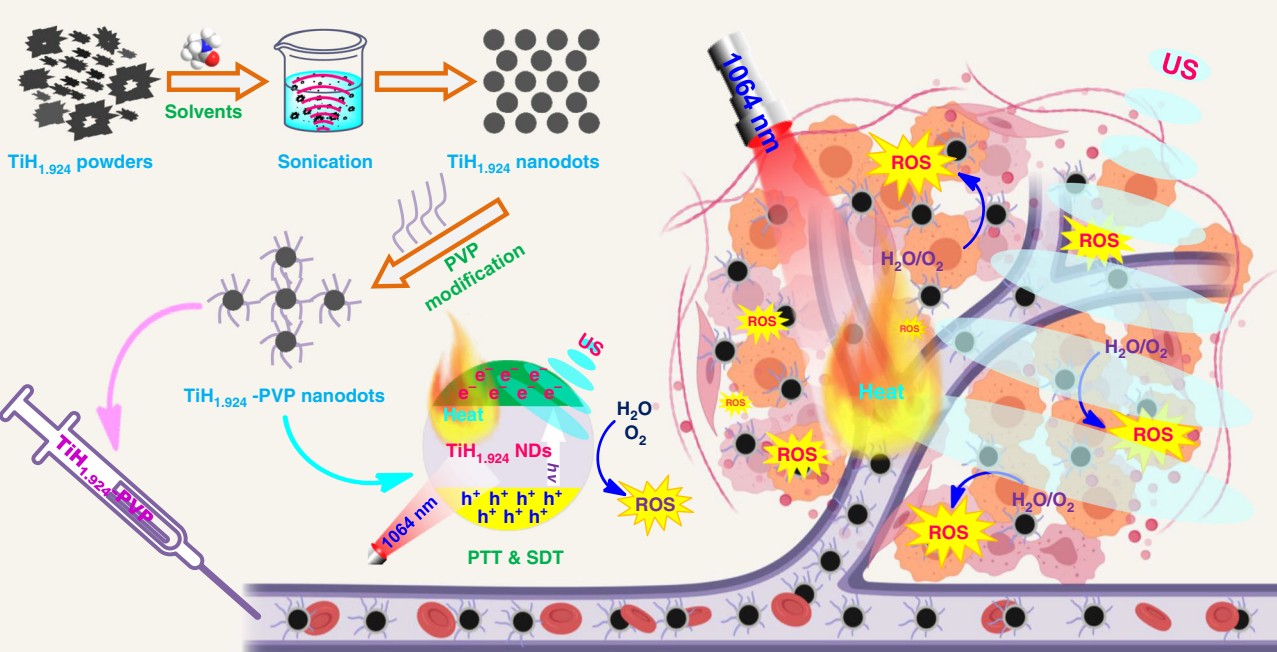

**Fig. 1 The preparation and application of $TiH_{1.924}$ nanodot.** Schematic illustration to show the preparation of $TiH_{1.924}$ nanodots by liquid-phase exfoliation and their applications for combined photothermal-sonodynamic cancer therapy.

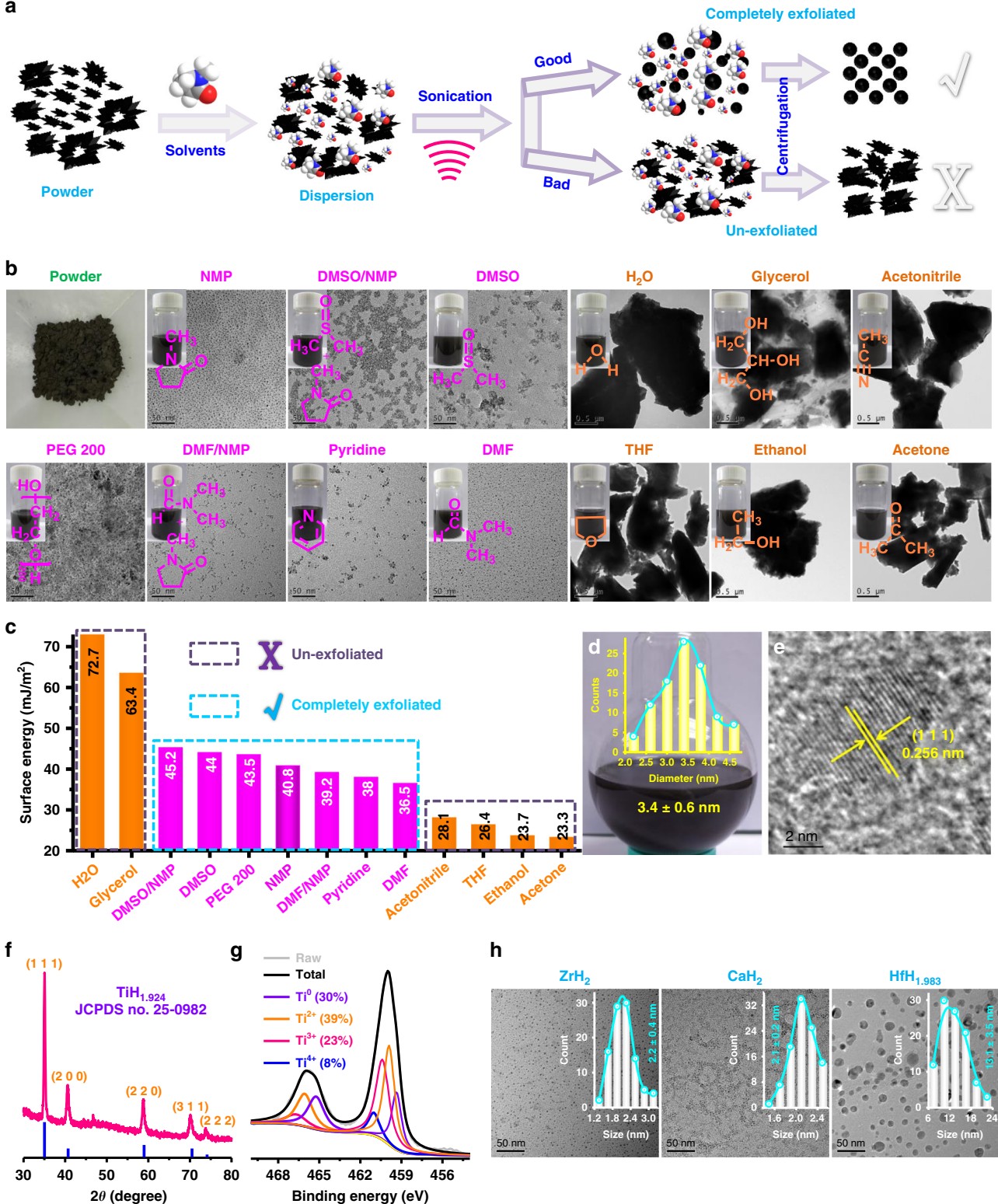

**Fig. 2 Preparation and characterization of TiH$_{1.924}$ nanodots. a** Schematic illustration to show light-phase exfoliation to prepare TiH$_{1.924}$ nanodots. **b** A photograph of commercial TiH$_{1.924}$ powder, the TEM images and corresponding photographs of exfoliated dispersions using various solvents (H$_2$O, glycerol, dimethyl sulfoxide/$N$-methyl pyrrolidone (DMSO/NMP) DMSO/NMP, DMSO, polyethylene glycol 200 (PEG 200), NMP, $N,N$-dimethylformamide (DMF)/NMP, pyridine, DMF, acetonitrile, tetrahydrofuran (THF), ethanol, and acetone) for TiH$_{1.924}$ exfoliation. **c** The surface energies of various solvents used to exfoliate TiH$_{1.924}$. **d** A photograph of exfoliated TiH$_{1.924}$ nanodots in NMP. Inset is the particle-size distribution (PSD) of TiH$_{1.924}$ nanodots determined by the TEM image ($n = 100$ nanodots examined over TEM images). **e** High-resolution TEM (HRTEM) image of TiH$_{1.924}$ nanodots. **f** XRD spectra of TiH$_{1.924}$ nanodots. **g** XPS spectra to show Ti 2p peaks for the TiH$_{1.924}$ nanodots sample. **h** TEM images and PSD of ZrH$_2$ nanodots, CaH$_2$ nanodots, and HfH$_{1.983}$ nanoparticles exfoliated in NMP ($n = 100$ nanomaterials examined over TEM images). A representative image of three biological replicates from each group is shown in **b**, **e**.

26.3; ethanol, 23.7; acetone, 23.3 mJ m$^{-2}$), TiH$_{1.924}$ powder could not be effectively exfoliated. When the surface energy of the solvent (DMSO/NMP, 45.2; DMSO, 44; PEG 200, 43.5; NMP, 40.8; DMF/NMP, 39.2; pyridine, 38; DMF, 36.5 mJ m$^{-2}$) reaches a range of 41 ± 5 mJ m$^{-2}$, successful exfoliation of the TiH$_{1.924}$ powder into small nanoparticles could be achieved. Therefore, we proposed that the successful exfoliation might be owing to the matching of surface energy between the applied solvents and the TiH$_{1.924}$ powder.

Among these 13 solvents, NMP offered excellent exfoliation efficiency and the obtained TiH$_{1.924}$ nanodots showed very uniform sizes and morphology (Fig. 2b). Thus, we employed NMP as the representative solvent to investigate the liquid-phase exfoliation of TiH$_{1.924}$ powder. After exfoliation of TiH$_{1.924}$ powder by sonication in NMP for different periods of time (Supplementary Fig. 1A–C), we found that the intensities of the X-ray diffraction (XRD) characteristic peaks decreased significantly by a time-dependent manner (Supplementary Fig. 1D). Particularly, after 20 min of ultrasonication, the TiH$_{1.924}$ powder was entirely exfoliated into nanodots. Most importantly, this sample could be scaled-up to prepare large amounts of TiH$_{1.924}$ nanodots with high quality, and the obtained TiH$_{1.924}$ nanodots with an average diameter of 3.4 ± 0.6 nm could be well dispersed in NMP (Fig. 2d). The high-resolution TEM (transmission electron microscope) determined the lattice spacing to be 0.256 nm (Fig. 2e), which could be assigned to the (1 1 1) lattice plane of TiH$_{1.924}$ (JCPDS No. 25-0982) (Fig. 2f)[34]. The energy dispersive spectrometer (EDS) spectrum also confirmed the existence of Ti elements (Supplementary Fig. 2). Based on the X-ray photoelectron spectroscopy (XPS, Supplementary Fig. 3), Ti with various valence states including Ti$^0$ (30%), Ti$^{2+}$ (39%), Ti$^{3+}$ (23%), and Ti$^{4+}$ (8%) were found in the obtained TiH$_{1.924}$ nanodots (Fig. 2g)[24]. Apart from the TiH$_{1.924}$ powder, we also successfully exfoliated zirconium hydride (ZrH$_2$), calcium hydride (CaH$_2$), and hafnium hydride (HfH$_{1.983}$) powder using the liquid-phase exfoliation technology with the assistance of NMP (Fig. 2h, Supplementary Fig. 4). The obtained ZrH$_2$ nanodots (2.2 ± 0.4 nm), CaH$_2$ nanodots (2.1 ± 0.2 nm), and HfH$_{1.983}$ nanoparticles (13.1 ± 3.5 nm) all showed uniform morphology, suggesting that the liquid-phase exfoliation technology could be a simple and universal method to prepare various types of metal hydrides nanomaterials.

**Sonodynamic and photothermal performance of TiH$_{1.924}$ nanodots.** The special valence structure of TiH$_{1.924}$ nanodots indicates that they might be activated under US irradiation as a sono-sensitizer (Fig. 3a). Thus, to explore whether TiH$_{1.924}$ nanodots could enhance sono-catalysis, 1,3-diphenylisobenzofuran (DPBF), a reactive oxide species (ROS) probe, was employed to detect the ROS generation by US-activated TiH$_{1.924}$ nanodots. After mixing TiH$_{1.924}$ nanodots with the DPBF probe, the UV-vis absorption spectrum of the mixture was monitored after different periods of ultrasound (US) irradiation (Fig. 3b, Supplementary Fig. 5). Undergoing a series of US irradiation time, the intensity of DPBF characteristic absorption peak at 420 nm showed significant decrease, suggesting the quenching of probe by the generated ROS. Compared with commercial TiO$_2$ nanoparticles and untreated TiH$_{1.924}$ powder, the exfoliated TiH$_{1.924}$ nanodots exhibited a higher DPBF oxidation rate under the same US irradiation (Fig. 3c, Supplementary Fig. 6), indicating that TiH$_{1.924}$ nanodots could serve as a stronger sono-sensitizer than TiO$_2$. In addition, TiH$_{1.924}$ sono-sensitizers with higher exfoliation degrees showed better sonodynamic performance (Supplementary Fig. 7). Electron spin resonance (ESR) detection was also performed to compare the generated ROS

($^1O_2$, $\cdot O_2^-$, and $\cdot OH$) between TiH$_{1.924}$ and TiO$_2$ sono-sensitizer. (Fig. 3d, Supplementary Figs. 8 and 9). The characteristic peak intensities of the TiH$_{1.924}$ plus US showed a great increase than that of TiO$_2$, further demonstrating that TiH$_{1.924}$ nanodots could be activated to generate large amounts of ROS under US irradiation.

To understand the mechanism of sono-sensitization effect of TiH$_{1.924}$ nanodots, the optical absorbance spectra of solid TiH$_{1.924}$ nanodots and the commercial TiO$_2$ nanoparticles were measured (Fig. 3e). Based on the optical absorbance spectra and the kubelka-munk theory (Fig. 3f)[35,36], the optical bandgap of TiO$_2$ was calculated to be ~3.1 eV, which was consistent with the previous reports[37–39]. Interestingly, the bandgap of TiH$_{1.924}$ nanodots was determined to be ~2.7 eV, much lower than that of TiO$_2$. The bandgap is related to the required minimum energy to realize electron excitation[40,41]. Thus, the lower bandgap means easier activation and would result in more ROS generation under external stimuli[42,43]. Based on the above discussion, the possible mechanism is proposed in Fig. 3g. Under the US irradiation, the valence electron receives energy and could transit from the valence band (VB) to the conduction band (CB), resulting in the generation of the electron-hole pairs and excess energy, which are captured by surrounding O$_2$ and H$_2$O molecules to generate ROS (e.g., $^1O_2$, $\cdot O_2^-$, $\cdot OH$). With a lower bandgap compared to that of TiO$_2$, TiH$_{1.924}$ nanodots thus could be easier to be activated to produce more ROS under US irradiation, useful for applications in SDT.

We then studied the optical properties of the obtained TiH$_{1.924}$ nanodots. The as-made TiH$_{1.924}$ nanodots showed black color and strong optical absorbance, which appeared to be independent to the wavelength and was extended to the second NIR (NIR-II) region (Fig. 3h), in which light would have much higher tissue-penetrating capability in comparison to that in the NIR-I window[14,44]. On this ground, we presented that TiH$_{1.924}$ nanodots could use as a photothermal agent for effective NIR-II PTT (1064 nm). The extinction coefficient of TiH$_{1.924}$ at 1064 nm was tested to be ~10.27 L g$^{-1}$ cm$^{-1}$ (Supplementary Fig. 10), which was higher than that of black titania nanoparticles (B-TiO$_{2-x}$, 5.54 L g$^{-1}$ cm$^{-1}$)[14], traditional graphene oxide (GO, 3.6 L g$^{-1}$ cm$^{-1}$)[45], and carbon nanodots (CQs, 0.35 L g$^{-1}$ cm$^{-1}$)[46]. Then, the photothermal performance of the TiH$_{1.924}$ aqueous solution was further evaluated under 1064-nm NIR II laser. Significant concentration-dependent and laser-power-dependent photothermal heating effect was observed for these TiH$_{1.924}$ nanodots (Fig. 3i, Supplementary Fig. 11). The photothermal conversion efficiency ($\eta$) of TiH$_{1.924}$ nanodots was calculated to be ~58.6% (Fig. 3j, Supplementary Fig. 12), much higher than those of widely reported photothermal agents, like gold nanorods (21%)[47], copper selenide (Cu$_{2-x}$Se) nanocrystals (22%)[48], copper sulfide (Cu$_9$S$_5$) nanocrystals (25.7%)[49], and prussian blue (41.4%)[50]. In addition, there was almost no change of photothermal performance post five times ON/OFF laser cycles, showing the high photothermal stability of these TiH$_{1.924}$ nanodots (Fig. 3k).

In order to increase their stabilities in the physiological environment, as-made TiH$_{1.924}$ nanodots were modified with polyvinyl pyrrolidone (PVP), which could stabilize TiH$_{1.924}$ nanodots likely via the chelating-coordination between the O atoms of PVP and Ti atoms of TiH$_{1.924}$ (Supplementary Fig. 13)[44,51]. The amount of PVP coated on the surface of TiH$_{1.924}$ nanodots was measured by thermogravimetric analysis (TGA) to be ~26.4%. Unlike as-made TiH$_{1.924}$ nanodots which could be well dispersed in water but would aggregate in the presence of salt (e.g., in phosphate-buffered saline, PBS), TiH$_{1.924}$-PVP nanodots showed great dispersity in both water, PBS, and cell culture medium for a week. Notably, the photothermal and sonodynamic performance of TiH$_{1.924}$ nanodots did not change after surface modification with PVP or additional H$_2$O$_2$ treatments (Supplementary Figs. 14–16).

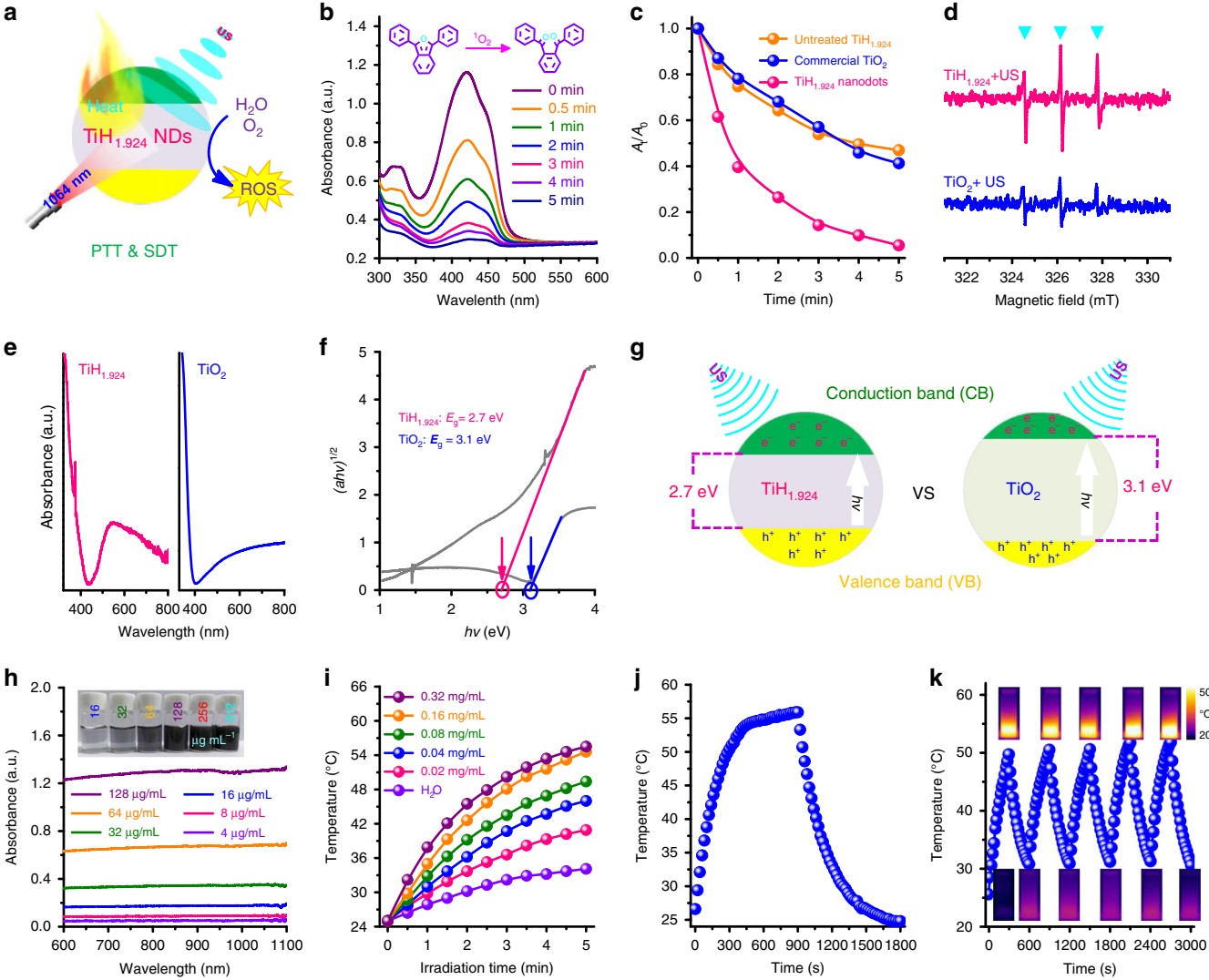

**Fig. 3 Sonodynamic and photothermal performance of TiH$_{1.924}$ nanodots. a** Schematic illustration of sonodynamic and photothermal properties of TiH$_{1.924}$ nanodots. **b** Time-dependent oxidation of DPBF indicating ROS generation by US-activated TiH$_{1.924}$ nanodots. **c** Comparison of DPBF oxidation by TiH$_{1.924}$ nanodots, untreated TiH$_{1.924}$, and commercial TiO$_2$ under US irradiation for 5 min. **d** ESR spectra demonstrating ROS ($^1$O$_2$) generation for TiH$_{1.924}$ and TiO$_2$ under US irradiation for 1 min. **e**, **f** Normalized absorption spectra (**e**) and optical bandgaps (**f**) of TiH$_{1.924}$ nanodots and TiO$_2$. **g** Schematic illustration of the activation mechanism of TiH$_{1.924}$ and TiO$_2$ under US irradiation. **h** UV-vis-NIR absorbance spectra at different concentrations of TiH$_{1.924}$ nanodots (4, 8, 16, 32, 64, and 128 μg mL$^{-1}$). The inset is the photograph of TiH$_{1.924}$ nanodots with different concentrations. **i** Concentration-dependent photothermal heating curves of TiH$_{1.924}$ nanodots (0, 0.02, 0.04, 0.08, 0.16, and 0.32 mg mL$^{-1}$). **j** The photothermal profile after laser exposure to reach a steady temperature and then to cool down by turning the laser off. **k** Heating/cooling profiles for five repeated ON-OFF cycles of laser irradiations.

**In vitro mild PTT-enhanced SDT.** With strong NIR-II absorbance and effective sono-sensitizing ability, we expected the utilization of TiH$_{1.924}$-PVP for synergistic photothermal-sonodynamic cancer therapy (Fig. 4a, Supplementary Fig. 17). Firstly, the standard methyl thiazolyl tetrazolium (MTT) assay demonstrated that TiH$_{1.924}$-PVP nanodots show negligible cytotoxicity even at high concentrations (400 μg mL$^{-1}$) toward 4T1 tumor cells (Fig. 4b). Next, the in vitro PTT-enhanced SDT induced by TiH$_{1.924}$-PVP was evaluated (Fig. 4c). After the mild PTT using the 1064-nm laser at the power density of 0.8 W cm$^{-2}$ for 10 min, the cell culture temperature increased to ~42 °C and the cell viability incubated with TiH$_{1.924}$-PVP showed a slight decrease (~80.9%). When further US irradiation was conducted (40 kHz, 3 W cm$^{-2}$, 1 min per cycle, 5 cycles), the 4T1 cell viabilities significantly decreased to ~10.6%, presenting increased cell killing compared to TiH$_{1.924}$-PVP treated cells exposed to US alone without pre-treatment by the NIR-II laser. In addition, the excellent cancer cell killing effect of mild PTT-enhanced SDT using TiH$_{1.924}$-PVP was also confirmed by live/dead co-staining (live cells, calcein-AM, AM; dead cells, propidium iodide, PI) (Fig. 4d). This increased SDT performance may be ascribed to the mechanism that the laser treatment could change the cell membrane permeability and enhance the cell uptake of TiH$_{1.924}$-PVP nanodots[52,53].

Next, 2,7-dichlorofluorescein diacetate (DCFH-DA, green color) and dihydroethidium (DHE, red color) staining assays were also performed to determine intracellular ROS generation and verify the mechanism of TiH$_{1.924}$-PVP as a sono-sensitizer to kill cancer cells under ultrasound (Fig. 4e, Supplementary Figs. 18 and 19)[54]. Cells in the control group, TiH$_{1.924}$-PVP only group, laser only group, US only group, laser/US group, and TiH$_{1.924}$-PVP/NIR group (mild PTT), all showed weak intracellular ROS-related

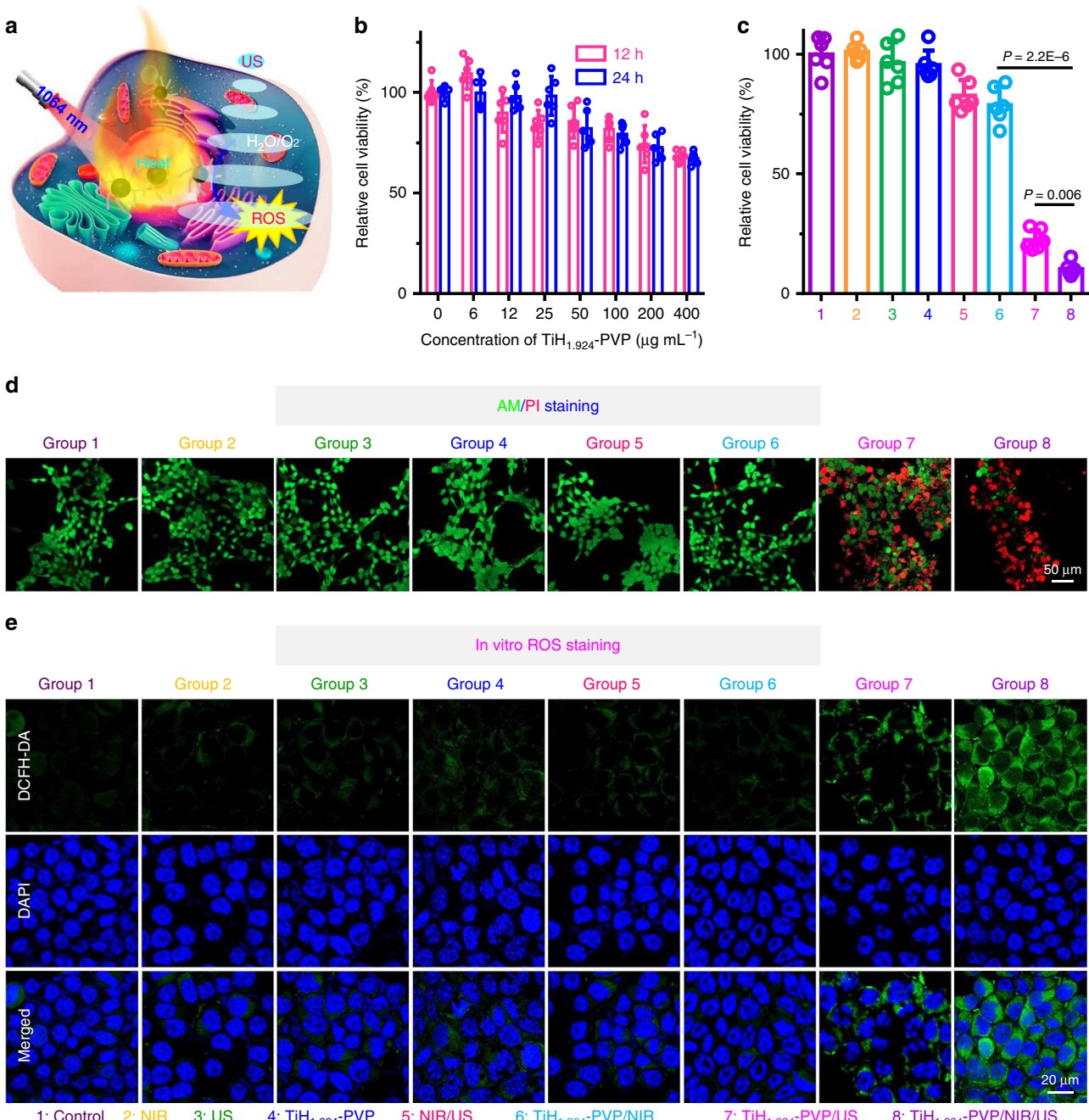

**Fig. 4 In vitro mild PTT-enhanced SDT via TiH$_{1.924}$-PVP. a** Schematic illustration of TiH$_{1.924}$-PVP for mild PTT-enhanced sonodynamic therapy. **b** Relative viabilities of 4T1 cells after incubation with various concentrations of TiH$_{1.924}$-PVP for 12 h and 24 h ($n = 6$ biologically independent samples). **c** Relative viabilities of 4T1 cells after different treatments, including control, TiH$_{1.924}$-PVP, NIR, US, NIR/US, TiH$_{1.924}$-PVP/NIR, TiH$_{1.924}$-PVP/US, and TiH$_{1.924}$-PVP/NIR/US ($n = 6$ biologically independent samples). **d** Confocal images of 4T1 cells stained with calcein AM (green, live cells) and propidium iodide (red, dead cells) after different treatments. **e** Confocal images of 4T1 cells stained with DCFH-DA after various treatments. The nuclei and intracellular ROS were stained by DAPI (blue) and DCFH-DA (green), respectively. TiH$_{1.924}$-PVP: 50 μg mL$^{-1}$, NIR laser: 1064 nm, 0.8 W cm$^{-2}$, 10 min, $T < 42\,°C$; US irradiation: 40 kHz, 3 W cm$^{-2}$, 1 min per cycle, 5 cycles. Data are presented as mean values ± SD. Statistical significance was calculated with two-tailed Student's $t$ test (**c**). A representative image of three biological replicates from each group is shown in **d**, **e**.

fluorescence. In contrast, strong fluorescent signals were clearly observed in cells from the TiH$_{1.924}$-PVP/US and TiH$_{1.924}$-PVP/NIR/US groups, demonstrating the effective intracellular ROS generation by TiH$_{1.924}$-PVP under US stimulation.

**Mild PTT-defeated tumor hypoxia.** After in vitro experiments, the in vivo behaviors of TiH$_{1.924}$-PVP were studied using

photoacoustic (PA) imaging and it could monitor the tumor uptake of NIR-absorbing TiH$_{1.924}$-PVP nanodots. After intravenous (i.v.) injection of TiH$_{1.924}$-PVP into 4T1-tumor-bearing balb/c mice for 8 h, much obvious PA signals were clearly appeared in the tumor site (Fig. 5a, b), verified the tumor uptake of TiH$_{1.924}$-PVP via the enhanced permeability and retention (EPR) effect. At the following time points, the PA signals

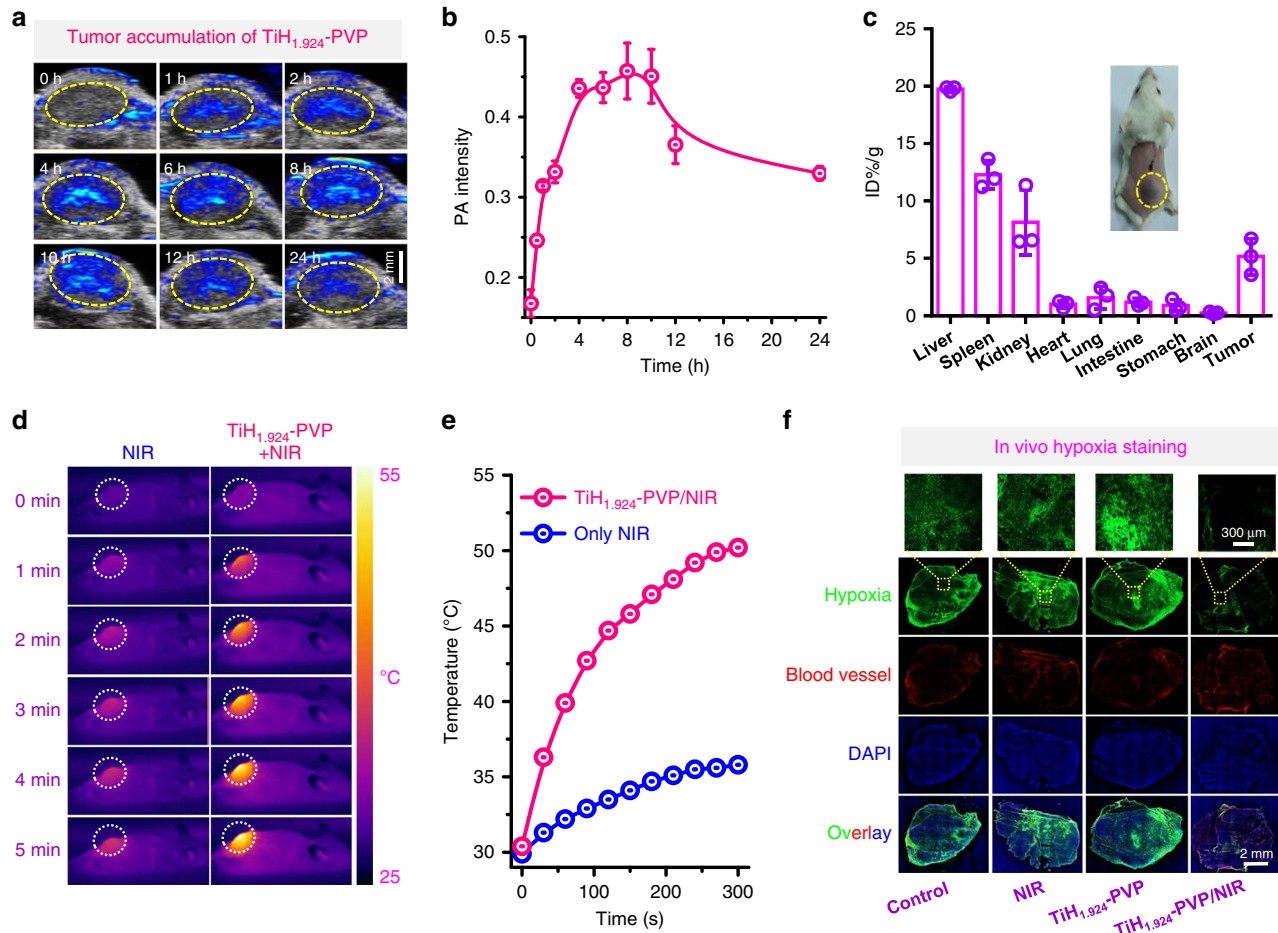

**Fig. 5 In vivo tumor accumulation and mild PTT-defeated tumor hypoxia via TiH$_{1.924}$-PVP. a** In vivo PA imaging of 4T1 tumor-bearing mice after intravenously injected with TiH$_{1.924}$-PVP. **b** Time-dependent tumor PA signals at 900 nm based on PA imaging data in **a** ($n = 3$ biologically independent mice). **c** Biodistribution of TiH$_{1.924}$-PVP in mice ($n = 3$ biologically independent mice). **d, e** IR thermal images (**d**) and temperature change curves (**e**) of 4T1 tumors under the 1064-nm laser irradiation, for untreated mice and TiH$_{1.924}$-PVP injected mice (irradiated at 8 h p.i.). **f** Representative immunofluorescence images of tumor slices after hypoxia staining. The nuclei, blood vessels, and hypoxia areas were stained by DAPI (blue), anti-CD31 antibody (red), and antipimonidazole antibody (green), respectively. TiH$_{1.924}$-PVP: 20 mg kg$^{-1}$; NIR laser: 1064 nm, 0.8 W cm$^{-2}$, 20 min. A representative image of three biological replicates from each group is shown in **f**. Data are presented as mean values ± SD.

gradually decrease, likely due to the clearance of the ultrasmall TiH$_{1.924}$-PVP from the tumor. In addition, the biodistribution of nanodots in the tumor was then quantitatively studied by measuring the content of titanium ions through inductively coupled plasma optical emission spectrometry (ICP-OES) at 8 h post injection (p.i.). The tumor uptake of TiH$_{1.924}$-PVP was determined to be ~5.2%ID g$^{-1}$, further confirming the efficient tumor accumulation of these nanodots (Fig. 5c).

Afterward, the in vivo photothermal performance of TiH$_{1.924}$-PVP for NIR II-induced hyperthermia was investigated own to the strong NIR absorbance and high tumor accumulation of TiH$_{1.924}$-PVP nanodots. And the surface temperature of tumors was recorded by infrared (IR) thermal imaging. 4T1 tumors-bearing mice post i.v. injection of TiH$_{1.924}$-PVP for 8 h were exposed to the 1064-nm laser irradiation (0.8 W cm$^{-2}$, 5 min) (Fig. 5d, e). Obviously, the tumor temperatures for TiH$_{1.924}$-PVP treated mice quickly increased to 50 °C, while that of the control group showed much less significant temperature increase.

Due to aberrant cancer cell proliferation and distorted blood tumor vessels, hypoxia arises in a wide variety of solid tumors and often causes the failure of cancer therapies, especially for those that consume oxygen in the cell killing process such as radiotherapy, photodynamic therapy (PDT), and SDT[55–57]. Based

on previous reports, the mild photothermal effect may increase intra-tumoral blood flow and then overcome tumor hypoxia[58–60]. To confirm this effect, immune-fluorescence hypoxia staining assay was conducted (Fig. 5f). Obviously, TiH$_{1.924}$-PVP plus NIR-II irradiation group showed a significantly decrease of the hypoxia signals, suggesting that the mild photothermal effect could efficiently overcome the tumor hypoxia, favorable for defeating hypoxia-associated SDT resistance.

**In vivo mild PTT-enhanced SDT.** Then, we conducted the mild PTT-enhanced SDT on 4T1 tumor-bearing mice using TiH$_{1.924}$-PVP. All of the mice were divided into five groups: (1) Control; (2) TiH$_{1.924}$-PVP (i.v. injection, 20 mg kg$^{-1}$); (3) NIR (1064 nm, 0.8 W cm$^{-2}$, 20 min, $T < 45$ °C) + US (40 kHz, 3 W cm$^{-2}$, 1 min per cycle, 20 cycles); (4) TiH$_{1.924}$-PVP + NIR; (5) TiH$_{1.924}$-PVP + US; (6) TiH$_{1.924}$-PVP + NIR + US. At 8 h post i.v. injection of TiH$_{1.924}$-PVP, the tumors were treated with 1064 nm laser and subsequent US irradiation (Fig. 6a). After various treatments, the tumor growth on different groups of mice was monitored. Compared with control, TiH$_{1.924}$-PVP injection alone, or NIR/US treated for saline injected mice showed no appreciable effect to the tumor group (Fig. 6b, c, Supplementary Fig. 20A). The mild PTT with TiH$_{1.924}$-PVP could only partially inhibit the tumor

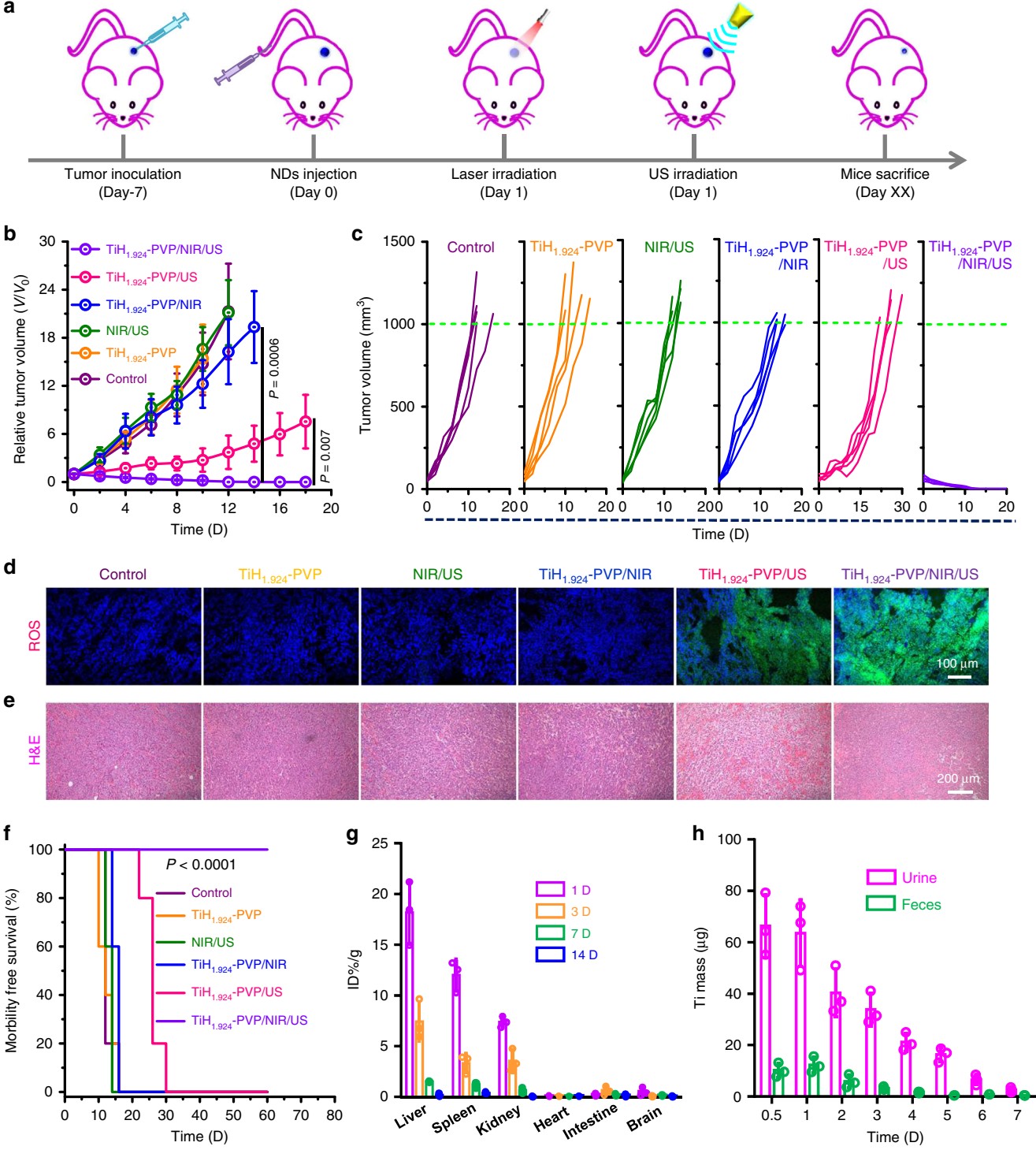

**Fig. 6 In vivo mild PTT-enhanced SDT via TiH$_{1.924}$-PVP. a** Schematic illustration to show the combination of PTT and SDT with TiH$_{1.924}$-PVP nanodots. **b, c** Average tumor growth curves (**b**) and individual tumor growth curves (**c**) on mice after different treatments, including control, TiH$_{1.924}$-PVP, NIR/US, TiH$_{1.924}$-PVP/NIR, TiH$_{1.924}$-PVP/US, and TiH$_{1.924}$-PVP/NIR/US ($n = 5$ biologically independent mice). **d** Micrograph of DCFH-DA stained tumor slices collected for mice receiving different treatments. **e** H&E stained tumor slices collected from different treatment groups. **f** Survival curves of mice after various treatments. **g** Biodistribution of TiH$_{1.924}$-PVP post i.v. injection in mice on different days ($n = 3$ biologically independent mice). **h** The detected Ti mass in urine and feces at different time points post i.v. injection of TiH$_{1.924}$-PVP ($n = 3$ biologically independent mice). The Ti contents were measured by ICP-OES. TiH$_{1.924}$-PVP: 20 mg kg$^{-1}$; NIR laser: 1064 nm, 0.8 W cm$^{-2}$, 20 min, $T < 45$ °C; US irradiation: 40 kHz, 3 W cm$^{-2}$, 1 min per cycle, 20 cycles. A representative image of three biological replicates from each group is shown in **d, e**. Data are presented as mean values ± SD. Statistical significance was calculated with two-tailed Student's $t$ test (**b**) and Logrank test (two-sided) for trend (**f**).

growth. The tumor growth of the SDT group (TiH$_{1.924}$-PVP/US) was remarkably suppressed, suggesting the excellent SDT performance of TiH$_{1.924}$-PVP. Interestingly, the TiH$_{1.924}$-PVP/NIR/US group showed the most remarkable therapeutic outcome, and the tumor tissues were completely eradiated without recurrence during two months. The survival time of mice in the SDT group (TiH$_{1.924}$-PVP/US) was prolonged compared to the other four groups (control, TiH$_{1.924}$-PVP, NIR/US, and mild PTT group) (Fig. 6f). More importantly, the mice in the mild PTT-enhanced SDT group showed 100% survival for two months, further demonstrating an obvious synergistic therapeutic outcome for the combined PTT-SDT with TiH$_{1.924}$-PVP in comparison to the single-modality SDT or mild PTT.

To further understand the mechanism of synergistic therapy, ROS staining of tumor slices was conducted to evaluate the ROS levels in the tumor post different treatments (Fig. 6d). Compared with the weak green fluorescence in tumor slices from control, TiH$_{1.924}$-PVP, NIR/US, and TiH$_{1.924}$-PVP/NIR groups, the TiH$_{1.924}$-PVP/US group showed obvious green fluorescence, and the strongest ROS-related fluorescence was observed in the combined laser plus US treatment group (TiH$_{1.924}$-PVP/NIR/US). Our results indicated the mild PTT could overcome tumor hypoxia and facilitate the SDT-triggered ROS production. In addition, hematoxylin and eosin (H&E) staining were conducted at 24 h after different treatments. Tumor cells were severely damaged in the SDT group and mild PTT-enhanced SDT group, while the other four groups showed little cell dead (Fig. 6e). These results confirmed the efficient synergistic effects induced by mild PTT-enhanced SDT, in the presence of TiH$_{1.924}$-PVP as a concurrent sono-sensitizer and photothermal nanoagent.

**The body clearance behaviors**. In two weeks after the treatment, the body weights of mice showed no significant change, indicating no apparent acute toxicity of TiH$_{1.924}$-PVP (Supplementary Fig. 20B). Next, we further investigated the body clearance behaviors of TiH$_{1.924}$-PVP after systemic injection, and a time-dependent biodistribution study was conducted after i.v. injection of TiH$_{1.924}$-PVP nanodots (Fig. 6g). Relatively high retention of TiH$_{1.924}$-PVP was observed in the liver (18.2 ± 3.1%ID g$^{-1}$), spleen (12.1 ± 1.5%ID g$^{-1}$), and kidney (7.4 ± 0.5%ID g$^{-1}$) at 24 h p.i. Importantly, rapid decrease of Ti levels in these organs was observed over time, indicating the efficient clearance of TiH$_{1.924}$-PVP. After 14 days, the Ti retention in major organs drastically decreased to be <0.5%ID g$^{-1}$, indicating the nearly complete clearance of TiH$_{1.924}$-PVP. To further investigate the clearance pathway, the Ti concentrations in the urine and feces were also measured. High levels of Ti were observed in the urine, strongly evidencing the elimination of TiH$_{1.924}$-PVP nanodots via the renal filtration pathway (Fig. 6h). In addition, H&E staining of the major organs (heart, liver, spleen, kidney, heart, lung, and brain) also confirmed the negligible toxicity of TiH$_{1.924}$-PVP to mice at this therapeutic dose (Supplementary Fig. 21). With efficient clearance and no acute toxicity, such TiH$_{1.924}$-PVP nanodots could be safe for in vivo use without long-term toxicity within the appropriate dose range.

## Discussion

In summary, nano-structured TiH$_{1.924}$ materials were synthesized via the liquid-phase exfoliation method. It was found that when the surface energy of the applied solvent had a good match with that of the TiH$_{1.924}$ powder, efficient exfoliation of such metal hydride powder into nanoparticles could be realized. Using the same method, a series of metal hydrides powders (TiH$_{1.924}$, ZrH$_2$, CaH$_2$, and HfH$_{1.983}$) were successfully exfoliated into small nanoparticles. With strong NIR-II absorbance and efficient

US-triggered ROS production ability, TiH$_{1.924}$ nanodots with PVP modification were further applied for the combined PTT-SDT therapy with great in vivo tumor destruction efficacy. Such TiH$_{1.924}$ nanodots present the following advantages as therapeutic nano-agent. (1) These TiH$_{1.924}$-PVP nanodots exhibit excellent sonodynamic performance in US-triggered ROS generation due to the reduced band-gap in TiH$_{1.924}$ compared to that of TiO$_2$. (2) The strong NIR absorption of TiH$_{1.924}$-PVP could enable enhanced photothermal-sonodynamic therapy, in which the mild hyperthermia-induced tumor hypoxia relief would lead to improved sonodynamic tumor killing. (3) Containing biocompatible elements (Ti and H), TiH$_{1.924}$-PVP nanodots with ultra-small sizes could allow their efficient body excretion without appreciable toxicity. Moreover, this work illustrates the promises of nano-structured metal hydrides nanomedicine platform against cancer and possibly other types of diseases.

## Methods

**Materials**. Titanium hydride (TiH$_{1.924}$), zirconium hydride (ZrH$_2$), calcium hydride (CaH$_2$), hafnium hydride (HfH$_{1.983}$), and N-methyl pyrrolidone (NMP) were purchased from Aladdin reagent Co., Ltd. (Shanghai, China). 1,3-diphenylisobenzofuran (DPBF), Polyvinyl pyrrolidone (PVP, MW 10 k), 2,2,6,6-tetramethylpiperidine (TEMP), and 5,5-dimethyl-pyrroline-N-oxide (DMPO) were obtained from Sigma-Aldrich. All chemicals were of analytical grade and used without further purification.

**Synthesis of TiH$_{1.924}$ nanodots**. 100 mg commercial TiH$_{1.924}$ powder was dispersed in 20 mL NMP. The mixture was treated under ultrasonication for different periods of time (Ultrasonic Cleaner, KQ-100KDB, power: ~100 W, temperature: ~15 °C). After ultrasonic treatment for 20 min, TiH$_{1.924}$ nanodots were obtained and further purified by centrifugation (64 k × g, 10 min) and washing repeatedly with anhydrous ethanol. Via the same method, zirconium hydride (ZrH$_2$) nanodots, calcium hydride (CaH$_2$) nanodots, and hafnium hydride (HfH$_{1.983}$) nanoparticles were also synthesized.

**Modification of TiH$_{1.924}$ nanodots**. The as-synthesized TiH$_{1.924}$ nanodots were modified by polyvinylpyrrolidone (PVP)[44,51]. Briefly, 20 mg TiH$_{1.924}$ and 200 mg PVP (MW 10 k) were dissolved in 50 mL anhydrous ethanol and refluxed at 50 °C for 8 h. After collecting by centrifugation (64 k × g, 10 min) and washing with water and ethanol, the final TiH$_{1.924}$-PVP nanodots were dispersed in deionized water, and stored at 4 °C for further use (concentration, 2 mg mL$^{-1}$).

**Characterization**. Transmission electron microscope (TEM) imaging and elemental mapping were carried out by FEI Tecnai F20 TEM. Powder X-ray diffraction (XRD) measurement was conducted by a PANalytical X-ray diffractometer equipped with CuKα radiation (λ = 0.15406 nm). XPS analysis was performed by the PHI Quantera SXM X-ray photoelectron spectrometer with an Al Ka monochromator source. ROS was quantified by an ESR spectrometer (Bruker EMXplus). UV-vis-NIR absorbance spectra were recorded by PerkinElmer Lambda 750 UV-vis-NIR spectrophotometer. The ultrasonic generator was made by Hainertec (Suzhou) Co., Ltd. The absolute Ti contents were determined by ICP-OES (inductively coupled plasma optical emission spectrometry).

**Quantitative analysis of the generation of ROS**. 1 mL TiH$_{1.924}$ (20 µg mL$^{-1}$) was mixed with 20 µL DPBF (1 mg mL$^{-1}$). After different US (40 kHz, 3 W cm$^{-2}$) durations, the absorbance changes of DPBF at 420 nm were recorded to quantify the generation of ROS by US-activated TiH$_{1.924}$. ESR technology combined with TEMP (for $^1O_2$ detection) or DMPO (for ·OH detection) was employed to detect different types of the generated ROS. In this case, 1 mL TiH$_{1.924}$ (20 µg mL$^{-1}$) was mixed with 20 µL TEMP (1 M) or 10 µL DMPO (1 M) and exposed to US irradiation (40 kHz, 3 W cm$^{-2}$) for 1 min. The characteristic peak signals were detected by the ESR spectrometer. The settings for the EPR spectrometer were as follows: center field, 3520 G; sweep width, 100 G; microwave frequency, 9.77 GHz; modulation frequency, 100 kHz; power, 20.00 mW.

**Photothermal performance of TiH$_{1.924}$ nanodots**. The photothermal performance of TiH$_{1.924}$ was analyzed by irradiating a glass cuvette containing a dispersion of TiH$_{1.924}$ nanodots. The extinction coefficient and the photothermal conversion efficiency were calculated according to the previous studies[44].

**Cellular experiments**. Murine breast cancer 4T1 cells were cultured in the standard cell culture medium at 37 °C under 5% CO$_2$. For the in vitro cytotoxicity test, 4T1 cells seeded in 96-well plates were incubated with different concentrations (0-400 µg mL$^{-1}$) of TiH$_{1.924}$-PVP for 12 h and 24 h. Relative cell viabilities were tested by the standard MTT assay.

For in vitro mild PTT-enhanced SDT, 4T1 cells were incubated with $TiH_{1.924}$-PVP (50 μg·mL$^{-1}$) for 8 h, followed by laser irradiation (1064 nm, 0.8 W cm$^{-2}$, 10 min, $T < 42\,°C$) or US irradiation (40 kHz, 3 W cm$^{-2}$, 1 min per cycle, 5 cycles). The cell viabilities were determined afterward by the MTT assay.

For live/dead staining, 4T1 cells under different treatments (including control, $TiH_{1.924}$-PVP, NIR, US, NIR/US, $TiH_{1.924}$-PVP/NIR, $TiH_{1.924}$-PVP/US and $TiH_{1.924}$-PVP/NIR/US) were stained with calcein AM (AM, live cell) and propidium iodide (PI, dead cell). For ROS detection, the treated 4T1 cells were stained with DCFH-DA (20 μM) for 30 min. All the images were acquired by a confocal laser scanning microscope (CLSM, Zeiss Axio-Imager LSM-800).

**Tumor model.** Balb/c mice were purchased from Nanjing Sikerui Biological Technology Co. Ltd, and all animal experiments were carried out under the permission from Soochow University Laboratory Animal Center. Six-week-old male Balb/c mice ($18 \pm 2$ g) were used as the animal model in this work. Mice were housed in groups of 5 mice per individually ventilated cage in a 12-h light–dark cycle (8:00–20:00 light; 20:00–8:00 dark), with constant room temperature ($21 \pm 1\,°C$) and relative humidity (40-70%). All mice had access to food and water ad libitum.

**Hypoxia tumor analysis.** For immunohistochemistry analysis, 4T1 tumor-bearing mice were intravenously injected with $TiH_{1.924}$-PVP (20 mg·kg$^{-1}$). At 8 h p.i., tumors on these mice were exposed to the 1064-nm laser irradiation for 20 min with their temperature maintained at ~45 °C. Then immediately, tumors were surgically excised for hypoxia staining assay using the Hypoxyprobe-1 plus kit (Hypoxyprobe Inc) following the standard protocol[61,62]. Anti-pimonidazole mouse monoclonal antibody conjugated with FITC (FITC-Mab1, Hypoxyprobe Inc.; Cat. No.: HP2-100Kit; Lot No.: 04-11-19; Clone: 4.3.11.3; Dilution: 1:200) and Alex 488-conjugated goat anti-mouse secondary antibody (Jackon Inc., Cat. No.: 115-545-003, Lot No.: 146108, RRID: AB_2338840; dilution: 1:200) for hypoxia staining. Rat anti-CD31 mouse monoclonal antibody (Biolegend Inc., Cat. No.: 102402, Lot No.: B226360, Clone: 390; dilution: 1:100) and Rhodamine-conjugated donkey anti-rat secondary antibody (Jackon Inc. Cat. No.: 712-025-150, Lot No.: 147079, RRID: AB_2340635; Dilution: 1:200) for blood vessel staining.

**In vivo mild PTT enhanced SDT.** Mice bearing 4T1 tumors (~100 cm$^3$) were divided into six groups ($n = 5$ per group): (1) control; (2) $TiH_{1.924}$-PVP (i.v. injection, 20 mg kg$^{-1}$); (3) NIR (1064 nm, 0.8 W cm$^{-2}$, 20 min, $T < 45\,°C$) + US (40 kHz, 3 W cm$^{-2}$, 1 min per cycle, 20 cycles); (4) $TiH_{1.924}$-PVP (i.v. injection, 20 mg kg$^{-1}$) + NIR (1064 nm, 0.8 W cm$^{-2}$, 20 min, $T < 45\,°C$); (5) $TiH_{1.924}$-PVP (i.v. injection, 20 mg kg$^{-1}$) + US (40 kHz, 3 W cm$^{-2}$, 1 min per cycle, 20 cycles); (6) $TiH_{1.924}$-PVP (i.v. injection, 20 mg kg$^{-1}$) + NIR (1064 nm, 0.8 W cm$^{-2}$, 20 min, $T < 45\,°C$) + US (40 kHz, 3 W cm$^{-2}$, 1 min per cycle, 20 cycles). At 8 h after i.v. injection, the tumors were treated with laser irradiation, or US irradiation, or laser irradiation + US exposure, sequentially. Tumor temperature and thermal images were monitored and recorded by an IR thermal camera (Infrared Cameras. Inc). Tumor sizes and body weight were monitored every two days. The tumor volumes were calculated by the formula: volume = length × width$^2$/2. For in vivo H&E staining, tumors in different groups were collected on the second day post treatment.

**In vivo metabolism study.** Healthy mice after i.v. injection with $TiH_{1.924}$-PVP (20 mg·kg$^{-1}$) was sacrificed at 1, 3, 7, and 14 days, respectively. The major organs were collected, with one halves used for H&E staining, and the other halves used for detection of Ti levels by ICP-OES after these organs were solubilized by aqua regia. To study the excretion pathway, mice after i.v. injection with $TiH_{1.924}$-PVP nanodots were kept in metabolic cages to collect their feces and urine at various time points, which were solubilized by aqua regia measured by ICP-OES to determine Ti levels.

**Software.** All statistical analyses were performed on Origin 8.5, Excel 2010 and GraphPad Prism 6. Fluorescent images were collected by Confocal Microscopy (Zeiss LSM 880) and analyzed by LAS AF Lite 3.2.0 Image J 1.74v. IR thermal images were collected by Infrared Camera (Fotric 255). Photoacoustic imaging data was processed by PA Tomography (Vevo LAZR). All other characterization of $TiH_{1.924}$ was conducted by these instruments as indicated in the Characterization section.

**Reporting summary.** Further information on research design is available in the Nature Research Reporting Summary linked to this article.

## Data availability
The authors declare that all data needed to evaluate the conclusion of this work are presented in the paper and the Supplementary Information. Other data related to this work are available from the corresponding authors upon reasonable request.

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

## Acknowledgements

This article was partially supported by the National Basic Research Programs of China (973 Program) (2016YFA0201200), the National Natural Science Foundation of China (51525203, 51761145041, 51572180), Collaborative Innovation Center of Suzhou Nano Science and Technology, a Jiangsu Natural Science Fund for Distinguished Young Scholars (BK20170063), and a Project Funded by the Priority Academic Program Development (PAPD) of Jiangsu Higher Education Institutions. L.C. was supported by the Tang Scholarship of Soochow University. In particular, we sincerely thank Hainertec (Suzhou) Co., Ltd. for providing the ultrasonic generator.

## Author contributions

Z.L. oversaw all research; Z.L., L.C., and F.G. designed the experiments; L.C. and F.G. synthesized the materials; F.G., N.Y., and Y.G. performed the sonodynamic and photothermal experiments; F.G., Y.N., and S.B. performed the cells experiments; F.G., X.W., M.C., and Q.C. performed animal experiments; Z.L., L.C., and F.G. wrote the paper; All authors reviewed and edited the paper.

## Competing interests

The authors declare no competing interests.
