## [Peer Review File · Nature Communications]

Reviewers' Comments:

Reviewer #1:

Remarks to the Author:

This paper describes a new 2D material for cancer therapy. It combines the properties of TiH_{1.924} nanodots as sono-sensitizers and as photothermal agents.

The results are very interesting and well presented. They cover several aspects spanning from the mechanisms of ROS production to the in vitro and in vivo anticancer activity. It deserves publication in Nature Communications, but I have some questions that the authors should address in their revised version.

DCFH-DA oxidation can be enzymatically catalyzed by the pair peroxidase/H₂O₂. Due to the existence of several substances that interfere with the formation of DCFH, this probe, when used in cellular systems, has better use as a marker of the cellular oxidative stress. So other ROS probes should be tested (see for example: Gomes et al. J Biochem. Biophys. Methods 2005, 65, 45-80).

The authors should better specify the ESR probes they used. It is not clear when TEMP or DMPO are used. Generation of singlet oxygen normally is generated via energy transfer mechanism, while in this case a charge transfer mechanism is proposed. The authors should clarify how singlet oxygen is generated with the appropriate probes.

The authors say that they quantify the ROS produced. However, no quantification data are reported.

Most of the characterizations seem to be performed in the material exfoliated into the organic solvents. This is not really clear, and should be specified in the legend of the figures. The exfoliated material has been dispersed in water for the biological test. This is not described clearly. Which are the conditions used? Centrifugation speed and time? Concentration of the stock solution?

Nanosized hydrides can be much more reactive than bulk material. The authors should discuss the stability of this material particularly in water.

How was the surface energy of the solvents measured? Are these data present in the literature? Is it possible to measure the surface energy of mixed solvent? I have a concern about this. Not sure it is possible.

How reliable is the XPS to measure the ratio of the different Ti valences? Deconvolution of the peak is generally not precise. The authors should at least provide averages performing the analysis a certain number of times on different nanodot samples.

The amount of PVP on the nanodots should be quantified.

The authors expressed the concentration of titanium hydride in molar concentration (mM). Without the molecular weight of the material how is this possible? It is more appropriate to express the amount as mass per volume ($\mu\text{g/ml}$).

Minor:

Scheme 1 is mentioned, but I think it is probably Figure 1;

"Solvent with different molecular structures...." has no meaning. Please modify

Please check the abbreviations as NMP is not abbreviated for example.

The superoxide O₂⁻ lacks of the radical.

Some of the figure panels are too small and it is difficult to see some text or the structures (see solvent structures, which are useless though).

Reviewer #2:

Remarks to the Author:

In this study, ultra-small titanium hydride (TiH_{1.924}) nanodots have been prepared as the potential sonosensitizer from its powder form via the liquid-phase exfoliation. The nanodots showed better sonocatalytic activity than reported TiO₂ NPs. The near IR absorption of the nanodots were used to produce mild photothermal heating to enhance intra-tumoral blood flow

and improve tumor oxygenation. Consequently, a remarkable synergistic therapeutic effect is achieved in the combined photothermal-sonodynamic therapy. This study is interesting, but I recommend to improve the quality of the manuscript by considering following comments.

1. Storage stability of the nanodots, obtained by exfoliation of TiH_{1.924} powder after PVP hydrophilization, is not provided. Thus, it is not sure whether these nanodots have similar or faster aggregation tendency, compared to hydrophilized TiO₂ nanoparticles.
2. The higher degree of exfoliation showed better sonocatalytic property, which might be mostly due to the increase in coordinately unsaturated Ti surface in the nanodots. What is the fate of the nanodots surface in H₂O₂-rich cancer environments? H₂O₂ has the ability to convert titanium hydride to oxygen deficient black TiO₂ at elevated temperature, which is a better sonocatalyst than TiO₂ nanoparticles. In cancer microenvironments, it is wondering if there is any formation of titanium oxide or oxohydrides in the nanodots surface. If H₂O₂-nanodots interaction occurs, is that influencing the sonocatalytic augmentation?
3. What is the mode of interaction between PVP and surface of Ti having variable oxidation state? how strong or stable the interaction is?
4. TiO₂ nanoparticles with highly reactive exposed (0 0 1) facets can be prepared from TiO₂ nanoparticles by similar exfoliation technique to enhance the photocatalytic activity. Does liquid-phase exfoliation make titanium hydride nanodots only having variable oxidation state or any reactive facet exposure, contributing to the sonocatalytic augmentation?

Itemized Responses to Reviewers' Comments

Reviewer #1: This paper describes a new 2D material for cancer therapy. It combines the properties of $TiH_{1.924}$ nanodots as sono-sensitizers and as photothermal agents. The results are very interesting and well presented. They cover several aspects spanning from the mechanisms of ROS production to the in vitro and in vivo anticancer activity. It deserves publication in Nature Communications, but I have some questions that the authors should address in their revised version.

1. DCFH-DA oxidation can be enzymatically catalyzed by the pair peroxidase/ H_2O_2 . Due to the existence of several substances that interfere with the formation of DCFH, this probe, when used in cellular systems, has better use as a marker of the cellular oxidative stress. So other ROS probes should be tested (see for example: Gomes et al. J Biochem. Biophys. Methods 2005, 65, 45-80).

Reply: We thank you for these important comments. According to the ROS staining results (Figure 4E), the very low DCFH signals were observed in the control groups, verifying the negligible effects of the peroxidase/ H_2O_2 for DCFH-DA oxidation. Thus, the strong DCFH signals in the SDT groups are mainly ascribed to the US-activated ROS generation by $TiH_{1.924}$ sono-sensitizer inside cancer cells. In addition, DCFH-DA was widely used as the probe to detect the intracellular ROS (e.g., Nat. Commun. 2018, 9: 1-14; Chem. Sci. 2018, 9: 2927-2933; J. Am. Chem. Soc., 2019, 141: 16243-16248.).

Based on your suggestion, we also selected dihydroethidium (DHE) as the ROS probe to evaluate the intracellular ROS generation for 4T1 cancer cells during the SDT treatment. As revealed in Figure S19, compared with control groups, obvious red fluorescent signals were also observed in these two SDT groups ($TiH_{1.924}$ -PVP/US and $TiH_{1.924}$ -PVP/NIR/US), indicating the efficient intracellular ROS generation by $TiH_{1.924}$ -PVP sono-sensitizer under US irradiation. We have added these results in the revised manuscript.

Supporting Figure S19. (A) Confocal images of 4T1 cells stained with dihydroethidium (DHE) probe after various treatments. (B) Quantitative analysis of intracellular red fluorescent signals in different groups as indicated.

Revision: Page 6, Paragraph 4, we have modified the following sentence.

“Next, 2, 7-dichlorofluorescein diacetate (DCFH-DA, green color) and dihydroethidium (DHE, red color) staining assays were also performed to determine intracellular ROS generation and verify the mechanism of $TiH_{1.924}$ -PVP as a sono-sensitizer to kill cancer cells under ultrasound (Figure 4E, Figure S18&19)⁵³.”

2. The authors should better specify the ESR probes they used. It is not clear when TEMP or DMPO are used. Generation of singlet oxygen normally is generated via energy transfer mechanism, while in this case a charge

transfer mechanism is proposed. The authors should clarify how singlet oxygen is generated with the appropriate probes.

Reply: Thanks for your suggestion. In the ESR detection, TEMP (2,2,6,6-tetramethylpiperidine) was used as a $^1\text{O}_2$ probe, while DMPO (5,5-dimethyl-pyrroline-N-oxide) was a $\bullet\text{OH}$ probe. In our system, it is possible that both energy transfer and charge transfer would occur when $\text{TiH}_{1.924}$ sono-catalyst is exposed to the ultrasound, leading to the generation of various types of ROS.

Revision: Page 11, Paragraph 2, we have modified the following sentence.

“1 mL $\text{TiH}_{1.924}$ ($20 \mu\text{g}\cdot\text{mL}^{-1}$) was mixed with 20 μL DPBF ($1 \text{mg}\cdot\text{mL}^{-1}$). After different US (40kHz , $3 \text{W}\cdot\text{cm}^{-2}$) durations, the absorbance changes of DPBF at 420 nm were recorded to quantify the generation of ROS by US-activated $\text{TiH}_{1.924}$. ESR technology combined with TEMP (for $^1\text{O}_2$ detection) or DMPO (for $\bullet\text{OH}$ detection) was employed to detect different types of generated ROS. In this case, 1 mL $\text{TiH}_{1.924}$ ($20 \mu\text{g}\cdot\text{mL}^{-1}$) was mixed with 20 μL TEMP (1 M) or 10 μL DMPO (1 M) and exposed to US irradiation (40kHz , $3 \text{W}\cdot\text{cm}^{-2}$) for 1 min.”

Revision: Page 5, Paragraph 2, we have modified the following sentence.

“Under the US irradiation, the valence electron receives energy and could transit from the valence band (VB) to the conduction band (CB), resulting in the generation of the electron-hole pairs and excess energy, which are captured by surrounding O_2 and H_2O molecules to generate ROS (e.g., $^1\text{O}_2$, $\bullet\text{O}_2^-$, $\bullet\text{OH}$).”

3. The authors say that they quantify the ROS produced. However, no quantification data are reported.

Reply: Thanks for your kind reminder. ESR could be used to quantitatively detect the generation of ROS ($^1\text{O}_2$ and $\bullet\text{OH}$). The peak intensities of ESR spectra indicated the ROS levels. As revealed in **Figure S8&S9**, the peak intensities of the $\text{TiH}_{1.924}$ plus US group were significantly higher than that in the TiO_2 plus US group, indicating that $\text{TiH}_{1.924}$ as a sono-sensitizer could generate more ROS under US irradiation. We have added the quantitative data in the **Figure S8&S9**.

Supporting Figure S8. Quantitative analysis of $^1\text{O}_2$ generation for these two groups based on Figure 3D.

Supporting Figure S9. (A) ESR spectra demonstrating ROS ($\bullet\text{OH}$) generation for $\text{TiH}_{1.924}$ and TiO_2 under US irradiation for 1 min. (B) Quantitative analysis of $\bullet\text{OH}$ generation for these two groups as indicated.

4. Most of the characterizations seem to be performed in the material exfoliated into the organic solvents. This is not really clear, and should be specified in the legend of the figures. The exfoliated material has been dispersed in water for the biological test. This is not described clearly. Which are the conditions used? Centrifugation speed and time? Concentration of the stock solution?

Reply: We thank you for your kind reminder. In this study, we initially investigated the exfoliation of $\text{TiH}_{1.924}$ powder in a number of common solvents. We found that N-methyl pyrrolidone (NMP) offered excellent exfoliation efficiency. After exfoliation, the obtained $\text{TiH}_{1.924}$ nanodots were purified by centrifuging (30k rpm, 10 min) and repeatedly washing with anhydrous ethanol. After PVP modification, the $\text{TiH}_{1.924}$ -PVP was dissolved in deionized water and stored at 4 °C for further use (concentration, 2 $\text{mg}\cdot\text{mL}^{-1}$). Before any biological test, the $\text{TiH}_{1.924}$ -PVP nanodots would be transferred into PBS buffers. We have added the related information in the revised manuscript.

Revision: Page 10, Paragraph 2, we have modified the following sentence.

“After ultrasonic treatment for 20 min, $\text{TiH}_{1.924}$ nanodots were obtained and further purified by centrifugation (30k rpm, 10 min) and washing repeatedly with anhydrous ethanol.”

Revision: Page 10, Paragraph 3, we have modified the following sentence.

“Briefly, 20 mg $\text{TiH}_{1.924}$ and 200 mg PVP (MW 10k) were dissolved in 50 mL anhydrous ethanol and refluxed at 50 °C for 8 h. After collecting by centrifugation (30k rpm, 10 min) and washing with water and ethanol, the final $\text{TiH}_{1.924}$ -PVP nanodots were dispersed in deionized water, and stored at 4 °C for further use (concentration, 2 $\text{mg}\cdot\text{mL}^{-1}$).”

5. Nanosized hydrides can be much more reactive than bulk material. The authors should discuss the stability of this material particularly in water.

Reply: Thanks for your important comments. The stability of $\text{TiH}_{1.924}$ nanodots dispersed in water was carefully investigated. As revealed in XRD spectra (**Figure R1**), after being stored in H_2O for long time, the black color of the samples was not changed, and the intensities of the characteristic peaks of our synthesized $\text{TiH}_{1.924}$ nanodots showed no obvious changes, suggesting the good stability of $\text{TiH}_{1.924}$ nanodots in water.

Figure R1. (A) The photographs of TiH_{1.924} nanodots after being stored in H₂O for 7 days. (B) XRD spectra of TiH_{1.924} nanodots after being stored in water for 1, 3, and 7 days.

6. How was the surface energy of the solvents measured? Are these data present in the literature? Is it possible to measure the surface energy of mixed solvent? I have a concern about this. Not sure it is possible.

Reply: We thank you for this good comment. In this study, the surface energy of these used solvents was referenced from previous studies (*Nano Lett.* 2015, 15, 5449-5454; *Science*, 2011, 331: 568-571). They used this parameter to investigate the liquid phase exfoliation of two-dimensional (2D) materials. The surface energy of these solvents was calculated based on the surface tensions of phases, the contact angles, and the Young-Laplace equation. The detailed information could be found in the above-mentioned literatures. In this work, we accidentally found that metal hydride powder could be efficiently exfoliated into nanoparticles by liquid phase exfoliation technology and the exfoliated results also followed this rule.

7. How reliable is the XPS to measure the ratio of the different Ti valences? Deconvolution of the peak is generally not precise. The authors should at least provide averages performing the analysis a certain number of times on different nanodot samples.

Reply: Thanks for your kind reminder. XPS measurements of the TiH_{1.924} nanodots samples (n=3) from different batches were carried out (**Figure R2**). The average ratio of the different Ti valences was estimated to be 30% (Ti⁰), 39% (Ti²⁺), 23% (Ti³⁺), and 8% (Ti⁴⁺).

Figure R2. (A) XPS spectra to show Ti 2p peaks for the different $\text{TiH}_{1.924}$ nanodots samples. (B) The ratios of the different Ti valences in these samples.

Figure 2G. XPS spectra to show Ti 2p peaks for the $\text{TiH}_{1.924}$ nanodots sample.

Revision: Page 4, Paragraph 1, we have modified the following sentence.

“Based on the X-ray photoelectron spectroscopy (XPS, Figure S3), Ti with various valence states including Ti^0 (30%), Ti^{2+} (39%), Ti^{3+} (23%), and Ti^{4+} (8%) were found in the obtained $\text{TiH}_{1.924}$ nanodots (Figure 2G)”

8. The amount of PVP on the nanodots should be quantified.

Reply: We thank you for this insightful comment. Thermogravimetric analysis (TGA) was carried out to determine the amount of PVP coating on the surface of the $\text{TiH}_{1.924}$ nanodots (Figure S13C). From the TGA, the amount of PVP coated on the surface of $\text{TiH}_{1.924}$ nanodots was determined to be ~26.4%. We have added this figure and some discussion in the manuscript.

Supporting Figure S13C. Thermogravimetric analysis (TGA) of the obtained $\text{TiH}_{1.924}$ before and after surface modification.

Revision: Page 6, Paragraph 2, we have added the following sentence.

“The amount of PVP coated on the surface of $\text{TiH}_{1.924}$ nanodots was measured by thermogravimetric analysis (TGA) to be ~26.4%.”

9. The authors expressed the concentration of titanium hydride in molar concentration (mM). Without the molecular weight of the material how is this possible? It is more appropriate to express the amount as mass per volume ($\mu\text{g/ml}$).

Reply: Thanks for your kind reminder. All of the “molar concentration (mM)” was changed into “mass concentration” in the revised manuscript.

Revision: Page 11, Paragraph 2, we have modified the following sentence.

“1 mL TiH_{1.924} (20 μg•mL⁻¹) was mixed with 20 μL DPBF (1 mg•mL⁻¹). After different US (40 kHz, 3 W•cm⁻²) durations, the absorbance changes of DPBF at 420 nm were recorded to quantify the generation of ROS by US-activated TiH_{1.924}. ESR technology combined with TEMP (for ¹O₂ detection) or DMPO (for •OH detection) was employed to detect different types of the generated ROS. In this case, 1 mL TiH_{1.924} (20 μg•mL⁻¹) was mixed with 20 μL TEMP (1 M) or 10 μL DMPO (1 M) and exposed to US irradiation (40 kHz, 3 W•cm⁻²) for 1 min.”

Revision: Page 6, Part of modification section, we have modified the following sentence.

“Briefly, 20 mg TiH_{1.924} and 200 mg PVP (MW 10k) were dissolved in 50 mL anhydrous ethanol and refluxed at 50 °C for 8 h.”

Minor:

10. Scheme 1 is mentioned, but I think it is probably Figure 1;

Reply: Thanks for your reminder. “Scheme 1” has been changed into “Figure 1”.

11. “Solvents with different molecular structures....” has no meaning. Please modify

Reply: Thanks for your reminder. We have modified the sentence as “Taking TiH_{1.924} for example, we initially sonicated commercial TiH_{1.924} powder in a number of exfoliation solvents.”

12. Please check the abbreviations as NMP is not abbreviated for example.

Reply: Thanks for your reminder. We have carefully checked the abbreviations in the manuscript.

13. The superoxide O₂⁻ lacks of the radical.

Reply: Thanks for your reminder. The superoxide •O₂⁻ has been involved.

14. Some of the figure panels are too small and it is difficult to see some text or the structures (see solvent structures, which are useless though).

Reply: Thanks for your reminder. We have modified the Figure 2 to be clearly seen.

Figure 2B. A photograph of commercial TiH_{1.924} powder, the TEM images and corresponding photographs of

exfoliated dispersions using various solvents (H₂O, glycerol, DMSO/NMP, DMSO, PEG200, NMP, DMF/NMP, pyridine, DMF, acetonitrile, THF, ethanol, and acetone) for TiH_{1.924} exfoliation.

Figure 2D. A photograph of exfoliated TiH_{1.924} nanodots in NMP. Inset is the particle-size distribution (PSD) of TiH_{1.924} nanodots determined by the TEM image.

Figure 2H. TEM images and PSD of ZrH₂ nanodots, CaH₂ nanodots and HfH_{1.983} nanoparticles exfoliated in NMP.

 Reviewer #2: In this study, ultra-small titanium hydride (TiH_{1.924}) nanodots have been prepared as the potential sonosensitizer from its powder form via the liquid-phase exfoliation. The nanodots showed better sonocatalytic activity than reported TiO₂ NPs. The near IR absorption of the nanodots was used to produce mild photothermal heating to enhance intra-tumoral blood flow and improve tumor oxygenation. Consequently, a remarkable synergistic therapeutic effect is achieved in the combined photothermal-sonodynamic therapy. This study is interesting, but I recommend improving the quality of the manuscript by considering following comments.

1. Storage stability of the nanodots, obtained by exfoliation of TiH_{1.924} powder after PVP hydrophilization, is not provided. Thus, it is not sure whether these nanodots have similar or faster aggregation tendency, compared to hydrophilized TiO₂ nanoparticles.

Reply: We thank you for this suggestion. The storage stability of TiH_{1.924}-PVP has been investigated in different physiological solutions, including H₂O, PBS, and RPMI 1640 cell culture medium for 1, 3, and 7 days. As revealed in **Figure S13D**, the TiH_{1.924}-PVP nanodots in different buffers remained largely constant without any aggregation over 7 days, indicating the good stability of them in physiological environments.

Supporting Figure S13D. The photographs of TiH_{1.924}-PVP in different buffers including H₂O, PBS, and RPMI 1640 cell culture medium for 1, 3 and 7 days.

Revision: Page 6, Paragraph 2, we have modified the following sentence.

“Unlike as-made TiH_{1.924} nanodots which could be well dispersed in water but would aggregate in the presence of salt (e.g. in phosphate buffered saline, PBS), TiH_{1.924}-PVP nanodots showed great dispersity in both water, PBS, and cell culture medium for at least a week.”

2. *The higher degree of exfoliation showed better sonocatalytic property, which might be mostly due to the increase in coordinately unsaturated Ti surface in the nanodots. What is the fate of the nanodots surface in H₂O₂-rich cancer environments? H₂O₂ has the ability to convert titanium hydride to oxygen deficient black TiO₂ at elevated temperature, which is a better sonocatalyst than TiO₂ nanoparticles. In cancer microenvironments, it is wondering if there is any formation of titanium oxide or oxohydrides in the nanodots surface. If H₂O₂-nanodots interaction occurs, is that influencing the sonocatalytic augmentation?*

Reply: Thank you for your insightful comments. The excellent SDT performance of TiH_{1.924} nanodots is mainly ascribed to the exposure of more catalytic reactive sites after the higher degree of exfoliation treatments. Due to the high level of H₂O₂ inside the tumor, we then investigated that whether the H₂O₂ inside the tumor would oxidize the TiH_{1.924} nanodots. The TiH_{1.924} nanodots have been dispersed in H₂O₂ solution (1 mM, that higher than the level inside tumor) for 7 days. XRD analysis was then carried out to find whether there has the formation of titanium oxide or ox-hydrides in the nanodots surface. Compared with untreated TiH_{1.924} nanodots (**Figure S16A**), the color of the sample and the characteristic peaks of the H₂O₂-treated TiH_{1.924} nanodots showed no obvious change, suggesting that H₂O₂ inside the tumor would have negligible effects on TiH_{1.924} nanodots. In addition, the sonodynamic performance of TiH_{1.924} nanodots did not change after H₂O₂ treatment (**Figure S16B-D**).

Supporting Figure S16. (A&B) The photographs (A) and XRD spectra (B) of $\text{TiH}_{1.924}$ nanodots after treated with H_2O_2 (1 mM) for 7 days. (C&D) Time-dependent oxidation of DPBF by US-activated $\text{TiH}_{1.924}$ nanodots (C) and H_2O_2 -treated $\text{TiH}_{1.924}$ nanodots (D). (E) Comparison of DPBF oxidation by untreated $\text{TiH}_{1.924}$ and H_2O_2 -treated $\text{TiH}_{1.924}$ under US irradiation.

Revision: Page 6, Paragraph 2, we have modified the following sentence.

“Notably, the photothermal and sonodynamic performance of $\text{TiH}_{1.924}$ nanodots did not change after surface modification with PVP or additional H_2O_2 treatments (Figure S14-S16).”

3. What is the mode of interaction between PVP and surface of Ti having variable oxidation state? How strong or stable is?

Reply: Thanks for your comments. PVP is a polymer widely used as a pharmaceutical excipient to suspend different types of insoluble drugs. Moreover, in many published reports, PVP has been used to modify various types of metal-containing nanoparticles via chelating-coordination between the O atoms of PVP and metal atoms (*Adv. Funct. Mater.*, 2019, 29: 1901722; *J. Am. Chem. Soc.*, 2017, 139: 16235-16247; *ACS Nano*, 2017, 11, 9467-9480). Therefore, the chelating-coordination between the O atoms of PVP and Ti atoms of $\text{TiH}_{1.924}$ nanodots may also occur in our system. After PVP modification, the $\text{TiH}_{1.924}$ -PVP nanodots showed good dispersion stability in the physiological environment.

As revealed in **Figure S13**, unlike as-made $\text{TiH}_{1.924}$ nanodots which could be well dispersed in water but would aggregate in the presence of PBS, $\text{TiH}_{1.924}$ -PVP nanodots showed great dispersity in H_2O , PBS and RPMI 1640 cell culture medium for a week, suggesting the good stability of the interaction between PVP and $\text{TiH}_{1.924}$ nanodots.

Supporting Figure S13. (A) The photographs of TiH_{1.924} and TiH_{1.924}-PVP dispersed in the H₂O and PBS. (B) Fourier transforms infrared spectrometry (FTIR) spectra of TiH_{1.924} and TiH_{1.924}-PVP samples. (C) Thermogravimetric analysis (TGA) of the obtained TiH_{1.924} before and after surface modification. (D) The photographs of TiH_{1.924}-PVP in different buffers including H₂O, PBS, and RPMI 1640 cell culture medium for 1, 3, and 7 days.

Revision: Page 6, Paragraph 2, we have modified the following sentence.

“In order to increase their stabilities in the physiological environment, as-made TiH_{1.924} nanodots were modified with polyvinyl pyrrolidone (PVP), which could stabilize TiH_{1.924} nanodots likely via the chelating-coordination between the O atoms of PVP and Ti atoms of TiH_{1.924} (Figure S13).”

4. TiO₂ nanoparticles with highly reactive exposed (0 0 1) facets can be prepared from TiO₂ powders by similar exfoliation technique to enhance the photocatalytic activity. Does liquid-phase exfoliation make titanium hydride nanodots only having variable oxidation state or any reactive facet exposure, contributing to the sonocatalytic augmentation?

Reply: We thank you for this insightful comment. Based on the XRD spectra, after being treated by liquid-phase exfoliation, the intensities of XRD characteristic peaks decreased significantly without new characteristic peak appeared, indicating no new lattice plane generation (**Figure R3**). The sono-catalytic augmentation is mainly ascribed to the exposure of more catalytic reactive sites after the exfoliation treatments. In addition, due to the micro-size of TiH_{1.924} powder, it is hard to achieve the bio-application of TiH_{1.924}. In this study, we unexpectedly found that TiH_{1.924} powder could be easily exfoliated into nanodots using the liquid-phase exfoliation technology, which would be critical to fabricate metal hydride nanomaterials suitable for potential biomedical applications.

By the ultrasound treatment of TiH_{1.924} powder, the more catalytic reactive sites of TiH_{1.924} nanodots would be exposed to further amplify the sonodynamic effects. With the effective sono-sensitizing ability and strong NIR-II

absorbance, the as-made $\text{TiH}_{1.924}$ nanodots could be utilized for synergistic photothermal-sonodynamic cancer therapy. Moreover, the $\text{TiH}_{1.924}$ nanodots due to the ultrasmall size could also be efficiently cleared out from the mouse body, showing no long-term toxicity. All of these unique properties were ascribed to the successful exfoliation of $\text{TiH}_{1.924}$ powder via the liquid-phase exfoliation technology.

Figure R3. XRD spectra of $\text{TiH}_{1.924}$ samples before and after liquid-phase exfoliation.

We thank the two reviewers for his/her insightful comments and suggestions, which greatly helped to improve the quality of our manuscript!

Reviewers' Comments:

Reviewer #1:

Remarks to the Author:

The authors carefully addressed all concerns raised by the referees. The manuscript has improved a lot and I suggest to publish it in the current form.

Reviewer #2:

Remarks to the Author:

The revised version is well prepared by reflecting the reviewers' comments. It can be published as it is.